# Effect of Crystallinity on the Field Emission Characteristics of Carbon Nanotube Grown on W-Co Bimetallic Catalyst

**DOI:** 10.3390/nano14100819

**Published:** 2024-05-07

**Authors:** Qi Yao, Yiting Wu, Guichen Song, Zhaoyin Xu, Yanlin Ke, Runze Zhan, Jun Chen, Yu Zhang, Shaozhi Deng

**Affiliations:** State Key Laboratory of Optoelectronic Materials and Technologies, Guangdong Province Key Laboratory of Display Material and Technology, School of Electronics and Information Technology, Sun Yat-Sen University, Guangzhou 510275, China; yaoq27@mail2.sysu.edu.cn (Q.Y.); wuyt79@mail2.sysu.edu.cn (Y.W.); songgch@mail2.sysu.edu.cn (G.S.); xuzhy23@mail2.sysu.edu.cn (Z.X.); stskyl@mail.sysu.edu.cn (Y.K.); zhanrz3@mail.sysu.edu.cn (R.Z.); stscjun@mail.sysu.edu.cn (J.C.); stsdsz@mail.sysu.edu.cn (S.D.)

**Keywords:** carbon nanotube, field emission, bimetallic catalyst, W-Co alloy

## Abstract

Carbon nanotube (CNT) is an excellent field emission material. However, uniformity and stability are the key issues hampering its device application. In this work, a bimetallic W-Co alloy was adopted as the catalyst of CNT in chemical vapor deposition process. The high melting point and stable crystal structure of W-Co helps to increase the grown CNT diameter uniformity and homogeneous crystal structure. High-crystallinity CNTs were grown on the W-Co bimetallic catalyst. Its field emission characteristics demonstrated a low turn-on field, high current density, stable current stability, and uniform emission distribution. The Fowler–Nordheim (FN) and Seppen–Katamuki (SK) analyses revealed that the CNT grown on the W-Co catalyst has a relatively low work function and high field enhancement factor. The high crystallinity and homogeneous crystal structure of CNT also reduce the body resistance and increase the emission current stability and maximum current. The result provides a way to synthesis a high-quality CNT field emitter, which will accelerate the development of cold cathode vacuum electronic device application.

## 1. Introduction

Carbon nanotube (CNT) has been proven as an excellent field emission material for a long time [1,2,3]. A lot of efforts have been put to applied CNT as a field emitter and electron source in electron microscope and microwave vacuum electronic devices, etc. [4,5,6]. However, approaching industrial application, it is rare to see CNT-based devices realized now. One of the problems hampering its application is its uniformity and stability issues. The method for CNT synthesis includes laser ablation, arc discharge, and chemical vapor deposition (CVD) [7]. Among them, CVD is the closest to an industrial manufacturing method at present, which has the advantages of simple operation, low temperature, high yield, good controllability, and green environmental protection. Based on the catalytic CVD method, the crystal structure of CNT strongly depends on the catalyst alloy and its catalytic properties. The diameter of CNT depends on the diameter of the catalyst nanoparticles. The chirality of CNT depends on the low symmetry structure of the catalyst [8,9,10].

As the traditional catalysts, Fe, Co, and Ni were popularly adopted to grow CNT. However, due to their relatively low melting point, their aggregated nanoparticles tend to form kinds of amorphous crystal structures. Thus, the CNTs grown on them also have different kinds of chirality, such as zigzag, armchair, and others. The diversity of the crystal structures of CNT includes different conductivities and field emission characteristics of CNT [11], further leading to the nonuniformity and instability of CNTs on a large scale [12]. Therefore, controlling the crystal structure homogeneity of the catalyst alloy is an important way to achieve uniform and stable field emission characteristics of CNT.

A high-melting-point bimetallic catalyst is one of the solutions to achieve stable and uniform crystallinity of catalytic nanoparticles. The catalyst metal (e.g., Fe, Co, or Ni) combined with the high-melting metal (e.g., W, Mo, or Pt) forms a compound bimetal alloy. Due to the interaction between their components, bimetallic catalysts have a higher melting point, a specific crystal structure, and stable catalytic performances superior to that of a monometallic catalyst. For the CNT synthesis, the bimetallic catalyst has several advantages, including high catalytic activity, lattice matching, and diameter control capability [13,14,15]. In the literature, several bimetals have been utilized to synthesize CNTs, such as Pt-Co and Mo-Co [16,17]. Among them, it was revealed that the W-Co alloy has a unique atomic arrangement that matches the selectively growth of specific CNTs structures, such as the chiral single-walled CNTs, with (12,6), (16,0), and (14,4) [9,18,19]. Moreover, the excess addition of W can help to stabilize the Co nanoparticle and avoid the Ostwald ripening effect [20], which has benefits for the control of the particle diameter during catalyst reduction process. Several researchers also focused on using the bimetallic catalyst nanoparticles to inhibit Ostwald ripening and synthesize CNTs with few defects and enhanced crystallinity [16,21].

It is interesting to know whether the CNT grown on a bimetallic catalyst can have stable and uniform field emission characteristics. In the literature, B.K. Singh prepared carbon nanotubes with the Mo-Fe bimetallic catalyst, which had a higher field enhancement factor and higher emission current density compared with the pure Fe catalyst [22]. Tang demonstrated that Pt-Ni-assisted MWNTs had lower turn-on fields of 2.0 V/μm and a threshold field of 3.5 V/μm as well as better stability [23]. However, more systematic works should be carried out to investigate the intrinsic mechanism of bimetallic catalysts impacting the CNT’s growth and its field emission characteristics.

In this work, we adopted W-Co as the bimetallic catalyst to grow highly uniform and crystallized CNTs. Raman and transmission electron microscope (TEM) were adopted to characterize the intermetallic phase of the alloy and the crystallinity of CNT. The field emission of single CNT and CNT film were both measured to evaluate the effect of the crystallinity of CNT grown on a bimetallic catalyst on its field emission uniformity and stability. The results provide a way to synthesis a high-quality CNT field emitter, which will promote the application development of cold cathode vacuum electronic devices.

## 2. Materials and Methods

To deposit W-Co bimetal catalyst alloys, magnetron ion sputtering (Gatan module 628) was adopted. Before W-Co deposition, a buffer layer of Al_2_O_3_ with thickness of 12 nm was first deposited on the silicon wafer. As a control group, the SiO_2_ buffer layer and bare Si wafer were also adopted as buffer layer to synthesis CNT, which resulted a low-quality and even bared CNT film. The reason was attributed to the nanoscale rough surface of Al_2_O_3_ layer, which can uniformly disperse the catalyst film to particles and restrict the aggregation of particles during the hydrogen reduction process. 

Then, W and Co were sputtered in sequence on the Al_2_O_3_ as the catalyst layer. In the literature, the bimetallic catalyst was synthesized as powder using chemosynthesis [24]. However, for electronic device application, thin-film deposition is the better method for large-scale CNT film synthesis. In this experiment, different thickness proportions of the W and Co layer were adjusted to determine the best combination of W-Co bimetal catalyst.

After that, a thermal CVD (TCVD) process was carried out to synthesize CNT. Under atmospheric pressure, the CVD tube was rapidly heated to the specified reduction temperature (700–800 °C) with an Ar flow of 1000 sccm. Upon reaching the reduction temperature, the substrate loaded with catalyst was imported to the tube with a hydrogen flow of 500 sccm. Then, ethylene (C_2_H_4_), used as the carbon source, was subsequently fed into the tube for CNT growth at a specified growth temperature (750–820 °C) for 1 min. Finally, the tube was cooled down to room temperature under a flow of hydrogen and argon.

In TEM characterization, the W-Co catalyst was directly sputtered onto a Cu grid and treated under the same reduction conditions, while CNTs were scratched down from the substrate and dispersed on the Cu grid. Structure and elemental mapping were performed using a 300 kV transmission electron microscope (TEM, Titan3 G2 60-300, FEI, Hillsboro, OR, USA), including energy dispersive X-ray analysis (EDX), elemental mapping in scanning transmission electron microscopy (EDX-STEM), and selected region electron diffraction (SAED) patterns. Raman spectroscopy measurements were conducted on the CNTs using an inVia Reflex confocal Raman spectrometer. The spot size was 1 μm, with a laser excitation wavelength of 532 nm and a laser power of 10 uW/μm^2^. The scanned range was kept between 100–3000 cm^−1^, and analysis was performed by fitting Lorentzian functions to the characteristic D and G peaks. 

Field emission characterization of CNT was measured in two ways: single CNT field emission test and CNT film test. In the single CNT field emission test, the CNT sample was placed in the SEM chamber. Once a CNT was selected for test, a tungsten needle tip (anode) was moved to the top of the CNT with a distance of 400 nm. Then, a voltage (0–200 V, Keithley 2612A digital source meter) was applied to drive on field emission, and the field emission current was recorded. In the CNT film test, a metal plate/phosphorous screen was used as the anode, and a ceramic piece (300 μm thick) was used as a space insulator placed between the anode and CNT cathode. Then, the voltage was applied on the anode to measure the field emission characteristics. The vacuum pressure of the chamber was around 1.0 × 10^−5^ Pa.

## 3. Results and Discussion

### 3.1. Synthesis and Charcterization of CNTs Grown on W-Co Catalyst

The CNTs grown on the W-Co bimetallic catalyst using the TCVD method at 800 °C were characterized and are shown in Figure 1. The SEM image (Figure 1a) illustrates the uniform and dense CNTs grown on the substrate. Raman characterization (Figure 1b) revealed a low I_D_/I_G_ ratio of 0.23, indicating a reduced presence of defects and a high crystallinity. The TEM (Figure 1c,d) image demonstrates that most CNTs possess diameters ranging from 4–5 nm, with higher magnification revealing a double-walled structure for these tubes. These characterizations collectively demonstrate that the few-walled CNTs with exceptional quality can be synthesized on the W-Co alloy.

The intermetallic phase of the catalyst nanoparticle formed between W and Co alloy is the main reason for the uniform and high crystallinity of CNT. In order to confirm the main catalytic phase in W-Co alloy, a W-Co film with the proportion of W:Co = 0.4:0.3 was sputtered and treated in the catalyst reduction process to form bimetal nanoparticle without continuing the CNT growth process. Then, it was taken out for characterization in TEM. The EDX-STEM elemental mapping (Figure 2a–d) clearly showed that the catalyst nanoparticle was formed, including W and Co elements, which both distributed uniformly in the particle. The high-resolution TEM image (Figure 2e) revealed that the nanoparticle is around 10.4 nm in diameter. Its lattice fringes have intervals of 0.203 nm, which is consistent with the (4 4 0) plane spacing of Co_2_W_4_C. The SAED analysis of its lattice structure (Figure 2f) identified the diffraction points corresponds to (2 0 0), (1 1 1), and (3 1 1) planes in the fast Fourier transform (FFT) pattern. According to the literature [25], compared to the similarities in structure and lattice constants by precise phase identification method, the most likely active bimetallic alloy phase in the W-Co catalyst was identified as Co_2_W_4_C for catalyzing the growth of CNT. 

In the W-Co combination, W acts as the high-melting metal to retain a stable crystal structure, while Co acts as the catalyst. The intermetallic phase formed between W and Co increases the crystal stability of Co. However, too many Co contents may not form the stable specific crystal structure, while too many W contents may reduce the catalyst particle density and further reduce the density of CNT. Thus, five different proportion of W-Co contents were synthesized to determine an optimal value. The thickness proportion of W vs. Co was sputtered on the substrate as 0.7 (nm):0.3 (nm), 0.6:0.3, 0.5:0.3, 0.4:0.3, and 0.3:0.3. Then, a TCVD recipe was carried out to grow CNT. The as-grown CNTs’ morphology, structure, and Raman spectrum are shown in Figure 3. In Raman spectrum, the ratio of the D-band peak (~1350 cm^−1^) to the G-band peak (~1580 cm^−1^) was used to evaluate the crystallinity of CNTs. A lower-intensity ratio (I_D_/I_G_) reflects a better CNT crystallinity. As shown in Figure 3k–o, the CNT grown at the proportion of W:Co = 0.4:0.3 has the lowest I_D_/I_G_ ratio (0.4), which indicates a better crystallinity. Also, the CNT grown at the proportion of W:Co = 0.6:0.3 has a similar ratio (0.5). The CNTs grown at the other proportions have much larger ratios (I_D_/I_G_ = 0.89 for W:Co = 0.3:0.3, I_D_/I_G_ = 0.76 for W:Co = 0.5:0.3, I_D_/I_G_ = 0.69 for W:Co = 0.7:0.3). The SEM (Figure 3a–e) and TEM (Figure 3f–j) images show the morphology and structure of CNT film. The CNT film grown at the proportion of W:Co = 0.4:0.3 has a dense and uniform CNT distribution, and most of the CNTs are double-walled. In comparison, the CNT film grown at W:Co = 0.6:0.3 formed several cumulations and clusters that are not uniformly distributed. And the layer number is around 3–4. The other CNTs with large ratios have a sparse CNT density, and the layer numbers are larger than ten. It is believed that insufficient W content cannot disperse and anchor the Co particle, which leads to an uncertain catalytic crystal phase and large catalyst particle; thus, the CNT has a large diameter, multiple wall numbers, and low crystallinity. On the contrary, excess W content decreases the activity of the Co nanoparticle, which may form amorphous carbon and carbon fiber [26]. Therefore, in our results, the optimized W-Co proportion was W:Co = 0.4:0.3 (molar ratio W:Co = 0.93).

To further increase the crystallinity of CNT, catalyst reduction temperature, carbon gas source flow, and CNT growth temperature were modified to investigate the influence of the synthesis parameters on the crystallinity of CNT. It was found that although the bimetal catalyst can withstand even higher temperature, higher reduction temperature and growth temperature are not always better. In the catalyst reduction process, the Ostwald ripening effect dominated the particle diameter at higher temperatures. Thus, the particles were inclined to aggregate as a large particle, which is not favorable for few-walled CNTs. In the CNT growth process, the C_2_H_4_ decomposed faster in the catalyst particle surface at higher temperatures, which may encapsulate the catalyst particle and reduce its catalytic activity. In our experiment, when the growth temperature was raised from the original 750 °C to 800 °C, the CNT density and crystallinity increased accordingly. When it was further increased to 820 °C, the CNT crystallinity decreased. Therefore, the reduction temperature and growth temperature were optimized to further increase the crystallinity of CNT using the W-Co bimetallic catalyst.

### 3.2. Field Emission Characterization of CNTs Grown on W-Co Catalyst

Then, five CNT samples with different I_D_/I_G_ values were synthesized, and their field emission characteristics were measured to evaluate the effect of crystallinity on the field emission characteristics of CNT using the W-Co catalyst. As shown in Figure 4, the Raman spectrum of the five CNT samples showed that they have different I_D_/I_G_ ratios, which are 0.23, 0.35, 0.40, 0.49, and 0.65. Table 1 shows the recipe parameters for the growth of CNT with different I_D_/I_G_.

Then, their field emission characteristics were measured in two ways: in situ single CNT test and CNT film test. In the in situ single CNT test, five CNTs samples with similar length and diameter were selected as the object of measurement through SEM observation. The field emission current voltage curve and the corresponding Fowler–Nordheim (FN) plot are shown in Figure 5a,d. It is obvious that the CNT with lower I_D_/I_G_ ratio has a lower turn-on field and larger maximum current. The turn-on field (defined as the field strength required to achieve a current density of 100 nA/cm^2^) of the CNT with I_D_/I_G_ = 0.23 was 140 V/μm, while that of the CNT with I_D_/I_G_ = 0.65 increased to 191 V/μm (Figure 5c). It was noted that the very large turn-on field is due to the nanometer scale gap (400 nm) between the anode and CNT. The gap and the diameter of CNT are in the same scale so that the top of the CNT cannot be treated as a tip but as a halfsphere [27]. In this model, the field enhancement effect is as small as less than one hundred. Thus, a stronger external field should be applied to drive on field emission. The maximum current (defined as the current that a single CNT can withstand before break down) of the CNT with I_D_/I_G_ = 0.23 was 10 μA (Figure 5d), corresponding to a current density of 5.7 × 10^6^ A/cm^2^. It is a very large value that decreased the theoretical emission current density. With the increase in I_D_/I_G_ value, the maximum current dropped to 4.5 μA, which is half of the best value. For this the reason, a low turn-on field is attributed to the reduced work function and the larger field enhancement factor in the high-crystallinity CNT. The maximum current is attributed to the small body resistance of high-crystallinity CNT. 

The work function and field enhancement factor are the key to its field emission characteristics, which both contribute on the FN plot, while the crystallinity of CNT also influences the work function. A smaller I_D_/I_G_ ratio means that CNTs have fewer defects and less amorphous carbon on the surface. The adhesion of amorphous carbon on the CNT surface leads to the decrease in the σ–π hybrid bond intensity, which makes the Fermi energy level move upward, resulting in a slight increase in work function [28]. The presence of defects in the CNT lead to inelastic electron tunneling, resulting in the energy loss of emitted electrons, thus exhibiting higher work function [29]. To determine separately the change of work function and field enhancement factor among the five samples, a Seppen–Katamuki (SK) chart was adopted to analyze the FN data. The SK chart is a diagram, of which abscissa and ordinate are the intercept and the slope of the FN plot [30], respectively. Figure 5d shows the FN plot obtained from the field emission current and voltage curve. The logarithm of the FN plot shows the relationship of [31]:(1)$$\log\left(\frac{I}{V^{2}}\right)=a+b\times \frac{1}{V}$$

The intercept is $a=\log\left(\frac{875\alpha \beta^{2}}{\varphi }\right)+4.26/\varphi^{1/2}$, and the slope is $b=\frac{-1.17\times 10^{3}\varphi^{\frac{3}{2}}}{\beta }$. There are three variables: β, φ, and α. α represents the effective emitting area, which is also related to the β and φ, following the relation of $\alpha =\alpha_{0}\exp(-9.74\times 10^{7}\gamma \beta \varphi^{\frac{1}{2}})$. In this calculation, $\alpha_{0}=1.77\times 10^{-12}$ cm^−2^; γ is the effect constant of emitter shape, according to the experimental date fitting, and γ = 5 × 10^−10^. Thus, β and φ decide the intercept and slope. Based on the FN formula, we can fit the FN plots of the five CNT samples. All the plots followed FN field emission behavior, showing a straight line. The fitted straight lines showed two categories of slope and intercept among the five samples. If we fix φ and change β, an equi-work function line can be drawn. In reverse, an equi-field enhancement factor line can also be drawn through the numerical calculation, as shown in Figure 5e. 

Then, the slope and intercept values of the five samples collected from the FN plot were drawn as five points in the SK chart. The change of β and Φ among the five samples can be seen in a clear way. The CNT with I_D_/I_G_ = 0.23 has a smaller work function of 4.6, while the other four CNTs have a similar work function of 4.9–5.1. The small I_D_/I_G_ means the CNT has few defects in the tube structure and little amorphous carbon on the tube surface, which both impact a decrease in the work function of CNT [32]. The CNT with I_D_/I_G_ = 0.23 also has a relatively high field enhancement factor of 73, while the CNT with I_D_/I_G_ = 0.65 has a low factor of 46. The field enhancement factor is related to the shape and morphology of the CNT, particularly the diameter and length of the CNT. The relatively low field enhancement factor is due to the nanoscale gap weakening the field enhancement effect, as discussed above. Based on the result, it is believed that the high-crystallinity CNT with small I_D_/I_G_ also has a smaller diameter and longer CNT. It is also proven that the bimetal catalyst would form uniform and small nanoparticles in an optimized CVD process. In all, the high-crystallinity of CNT grown on a bimetal catalyst has a lower work function and higher field enhancement factor, which both reduce the turn-on field.

As the second reason, the high crystallinity of CNT decreases the scattering of electrons when electron flow goes through the tube, thus resulting a small body resistance and less Joule heat generation [29]. Consequently, the voltage drop on the tube is small, which results in a low turn-on field. The temperature rise on the tube is small, which means a larger current can go through the tube and avoid burning out the tube, so the measured maximum current increases. In summary, the high crystallinity of CNT grown on bimetal catalyst has a relatively low work function and body resistance, which help to obtain a stable and uniform field emission characteristic.

The field emission characteristics of the five CNT films with different I_D_/I_G_ were also measured (Figure 6), which showed the same trend as single CNTs. The sample with I_D_/I_G_ = 0.23 had a lowest turn-on filed of 0.45 V/μm (defined as the field strength required to achieve a current density of 10 µA/cm^2^), while the one with I_D_/I_G_ = 0.65 had a larger turn-on field of 2 V/μm. In the corresponding FN plots (Figure 6b), the two samples with smaller I_D_/I_G_ had a gentler slope compared to the other three samples, which means the small I_D_/I_G_ samples had a low work function or high field enhancement factor. 

The largest field emission current density reached 6 mA/cm^2^ in the CNT film with I_D_/I_G_ = 0.23 and 0.36, while the one with I_D_/I_G_ = 0.49 and 0.65 had around 3 mA/cm^2^, which is half of the best value. Compared with the field emission characteristics of CNT prepared by bimetallic catalysts previously reported, Table 2 shows that the CNT grown on the W-Co catalyst has an emission current density 3–6 times higher than the other works. Its turn-on field is 4–6 times lower than the other works. Figure 6c shows the field emission current stability. The current fluctuation was defined as σ = (I_max_ − I_min_)/(I_max_ + I_min_). The sample with lowest I_D_/I_G_ has the best emission stability, with a current fluctuation of 5.0%. The sample with largest I_D_/I_G_ has the worst value of 9.6%. The maximum current density and current stability are strong related to the crystallinity of CNT. A highly-crystallinity CNT without defects reduces the body resistance and Joule heat generation, definitely improving the current stability and helping the CNT to reach a high current density. 

Figure 6d shows the emission site distribution of CNT films. The two samples with I_D_/I_G_ = 0.23 and 0.36 showed a dense emission site distributed uniformly on the whole sample area. In contrast, the other three samples showed a nonuniform site distribution. Some emission sites showed a particularly bright spot, and most of the sites were distributed at the edge of the sample. The reason is related to the diameter uniformity of CNT. A smaller-diameter CNT can obtain a large field enhancement effect and thus a lower turn-on field. Under the same applied electric field, the small-diameter CNTs take precedence to emit current over the larger-diameter CNTs. A uniform diameter distribution results in a uniform emission site distribution. The emission site distribution confirmed that the high-crystallinity CNT film grown on the bimetal catalyst has a uniform diameter.

## 4. Conclusions

A bimetallic catalyst W-Co was adopted to grow CNT in the TCVD process. The W-Co proportion was optimized as W:Co = 0.4:0.3 to form a stable intermetallic-phase Co_2_W_4_C catalyst nanoparticle with uniform particle diameter distribution. Based on the bimetallic catalyst, high-crystallinity CNTs with small I_D_/I_G_ were synthesized. The field emission characteristics showed that the high-crystallinity CNT grown on W-Co catalyst has excellent characteristics, including low turn-on field (0.45 V/μm), high current density (6 mA/cm^2^), stable current stability (5.0%), and uniform emission distribution. The analysis revealed that the CNT has a relatively low work function and high field enhancement factor. The high crystallinity and homogeneous crystal structure of the CNT also reduce the body resistance and increase the emission current stability and maximum current. The results demonstrated that CNT grown on the W-Co bimetallic catalyst is a promising field emission material that may solve the uniformity and stability issues in device application.

## Figures and Tables

**Figure 1 nanomaterials-14-00819-f001:**
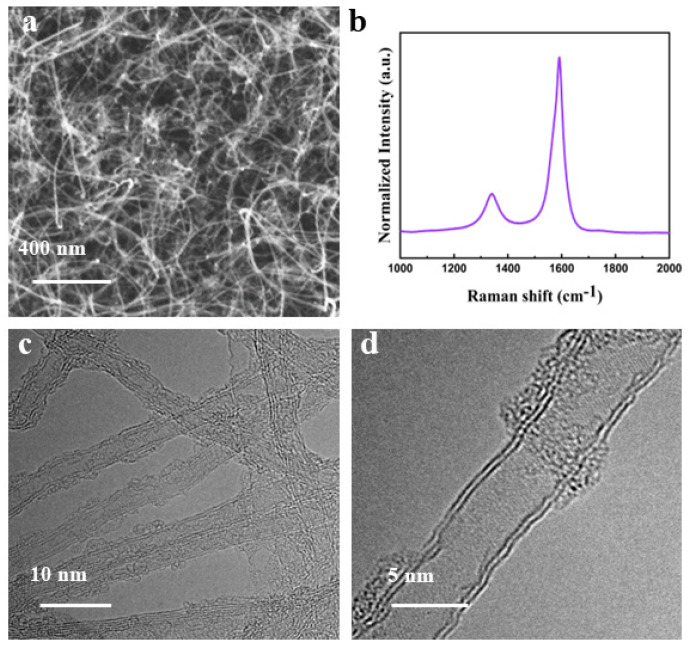
Characterization of high-quality CNTs grown from W-Co catalyst. (**a**) SEM image, (**b**) Raman spectra, and (**c**,**d**) TEM image.

**Figure 2 nanomaterials-14-00819-f002:**
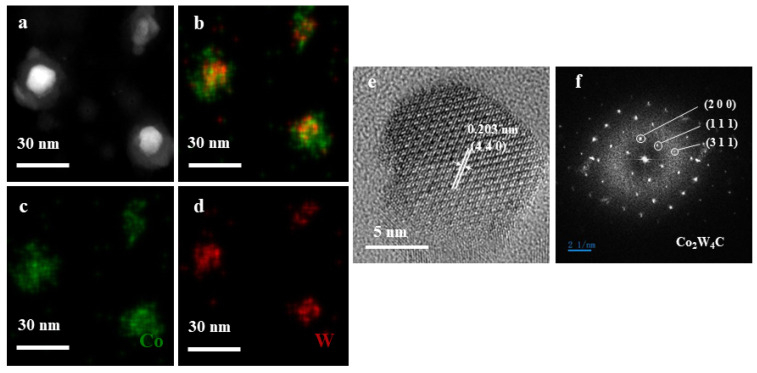
TEM and TEM characterization of W-Co-C nanocrystals after treatment with H_2_ at 750 °C: (**a**) HAADF image and (**b**–**d**) the corresponding elemental maps of the W-Co-C nanocrystals; (**e**) HRTEM image and (**f**) the corresponding fast Fourier transformation (FFT) patterns.

**Figure 3 nanomaterials-14-00819-f003:**
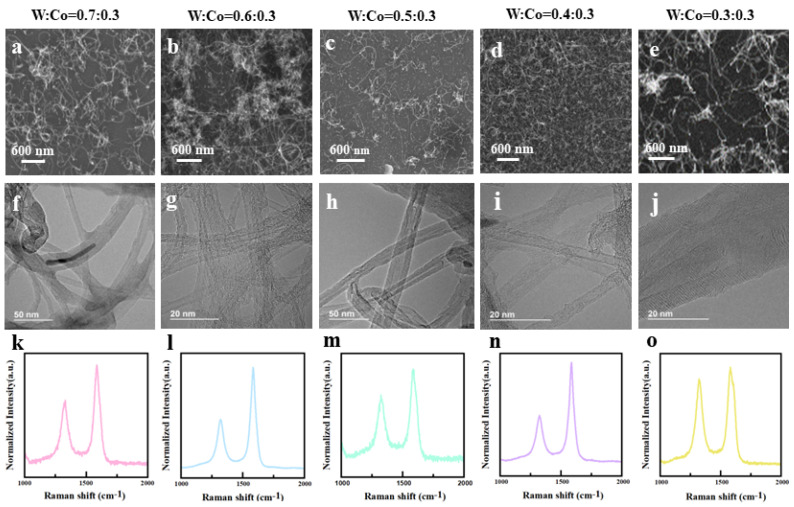
(**a**–**e**) SEM image of CNTs’ morphology, (**f**–**j**) TEM image of CNT’ structure, and (**k**–**o**) Raman spectrum of CNTs which were grown on the W-Co catalyst with different W-Co proportions, including: 0.7 (nm):0.3 (nm), 0.6:0.3, 0.5:0.3, 0.4:0.3, and 0.3:0.3.

**Figure 4 nanomaterials-14-00819-f004:**
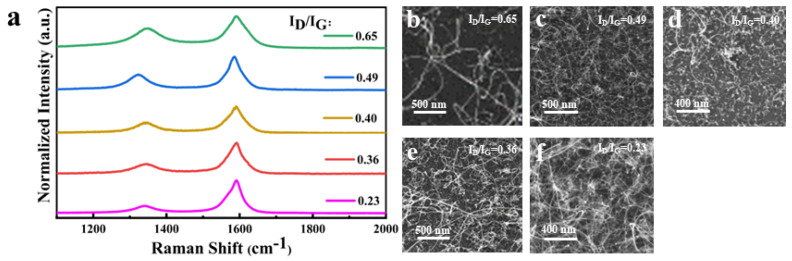
(**a**) Raman spectrum of five CNT samples using W-Co bimetal catalyst with different I_D_/I_G_ from 0.23 to 0.65 and (**b**–**f**) SEM image of CNT morphology.

**Figure 5 nanomaterials-14-00819-f005:**
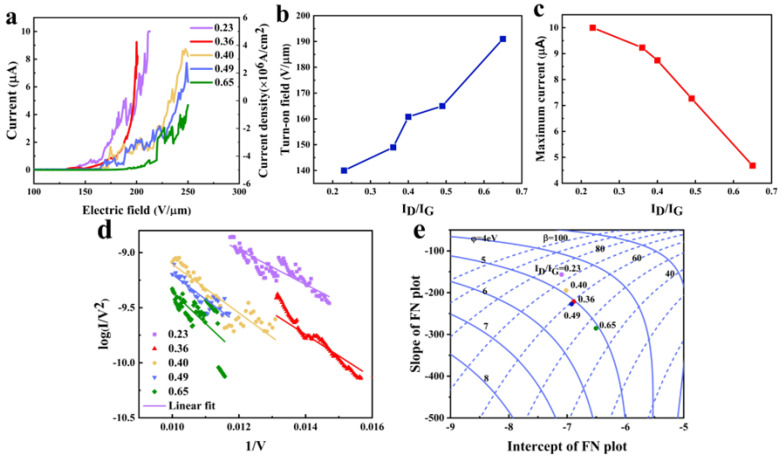
Field emission characteristics of single CNT with different I_D_/I_G_ from 0.23–0.69, (**a**) current vs. voltage curve, (**b**) relation of turn-on field with I_D_/I_G_, (**c**) relationship of maximum emission current with I_D_/I_G_, and (**d**) the corresponding FN plot and (**e**) SK chart.

**Figure 6 nanomaterials-14-00819-f006:**
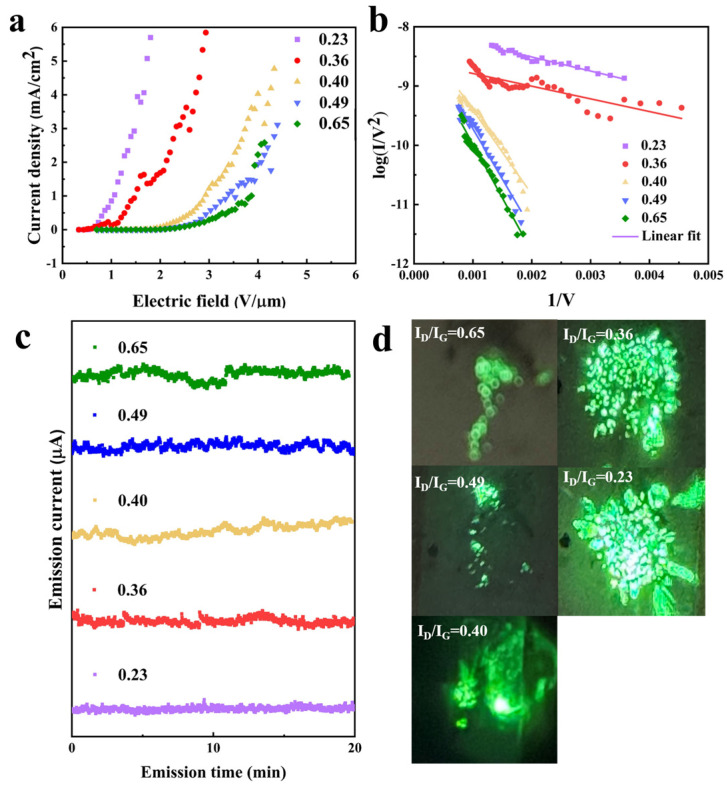
Field emission characteristics of CNT film with different I_D_/I_G_ from 0.23–0.69, (**a**) current vs. voltage curve, (**b**) the corresponding FN plot, (**c**) current stability, and (**d**) field emission site distribution.

**Table 1 nanomaterials-14-00819-t001:** Five CNT samples using W-Co = 0.4:0.3 catalyst with different growth process parameters.

I_D_/I_G_	Catalyst Treatment Temperature/°C	Catalyst Treatment Atmosphere	CNT Growth Temperature/°C	CNT Growth Atmosphere
0.23	750	Ar:H_2_ = 290:100 sccm	800	Ar:H_2_:C_2_H_4_ = 330:500:100 sccm
0.36	800	Ar:H_2_ = 290:100 sccm	800	Ar:H_2_:C_2_H_4_ = 330:500:100 sccm
0.40	800	Ar:H_2_ = 500 sccm	800	Ar:H_2_:C_2_H_4_ = 330:500:100 sccm
0.49	750	Ar:H_2_ = 290:100 sccm	750	Ar:H_2_:C_2_H_4_ = 330:500:100 sccm
0.65	750	Ar:H_2_ = 500 sccm	750	Ar:H_2_:C_2_H_4_ = 330:500:100 sccm

**Table 2 nanomaterials-14-00819-t002:** Comparison of field emission characteristics of CNT grown on bimetallic catalysts.

Catalyst	J_max_ (mA/cm^2^)	Turn-on Field (V/μm)	Reference
W-Co	6	0.45	This work
Fe-Mo	1.16	2	[22]
Co-Mo	1.14	3.17	[16]
Ni-Co	2.17	>2.5	[33]
Zr-Fe	\	3.2	[34]
Fe-Ti	\	13.9	[35]

This table turn-on electric field defined as the electric field at 10 µA/cm^2^.

## Data Availability

Data will be available upon request.

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
