# Peer review of "Effect of Crystallinity on the Field Emission Characteristics of Carbon Nanotube Grown on W-Co Bimetallic Catalyst"

_nanomaterials, 2024, doi:10.3390/nano14100819_

Round 1
Reviewer 1 Report
Comments and Suggestions for Authors
I have read the submitted original article entitled “Effect of crystallinity on the field emission characteristics of carbon nanotube grown on W-Co bimetallic catalyst” by Shaozhi Deng et al to Nanomaterials/MDPI. The reported carbon nanotubes (CNTs) have been applied and investigated in various areas this is mainly due to their unique characteristics of large aspect ratio, high electrical conductivity, and good thermal stability. The improvement of CNT film production with emission enhancement is crucial to promote the device’s developments. In the literature, there are reports on various catalyst materials such as tungsten, molybdenum, and sulfur could be used as auxiliary catalysts to improve catalyst activity and increase the yield of carbon nanotubes. Bimetallic catalysts such as Ni and Co have also been reported. In the submitted work, Den et al reported bimetallic W-Co alloy as catalysts in the CVD process. The submitted work falls in the significant area of research. The work has some new insights that deserve to be published in the chosen outlet. The manuscript is fairly well written.
The following points are to be considered during revision.
· In the abstract (line 16), acronyms such as “FN” and “SK” must be explained.
· In the section introduction, the advantages of using the CVD process must be detailed.
· Single-walled CNTs with diameters smaller than 1 nm were synthesized under low pressure (1 × 10−4 Pa) and low temperature (400 °C), showing that Pt has high catalytic activity. Being this is the case, why use a bimetallic catalyst?
· The observed Raman spectra could be correlated to the similar catalyst studies reported recently 10.1039/D3DT03736C.
· Whether supported bimetallic catalysts grow narrower CNTs with a higher growth density than the case of corresponding single metal catalysts.
· The Raman spectra of the five CNT samples are presented with Id/IG ratios but must be discussed how they influence the properties.
· Compare the J-E properties and benchmark them with the reported values.
· Is there any emission stability test undertaken?
· What is the role of bimetallic catalysts and importantly W in the presence of Co?
· What is the surface area of obtained CNT for W-CO proportion 0.4:0.3?
· The section conclusion must be qualitative having some metric performance.
Reviewer 2 Report
Comments and Suggestions for Authors
The paper deals with CNTs synthesis over bimetallic W-Co catalyst. The authors determined the optimal composition of the catalyst in terms of W:CO ratio, and the optimal temperature of reduction and the temperature of CNT growth. The results are interesting and the paper could be published, but it needs some revision. Firstly, the authors should read it throughout and correct all places where shortcuts were used, for example:
From line 150:
"at the proportion of 0.4:0.3" - proportion of W to CO, please add.
"large ratios" - is very general, please specify.
"film grown at 0.6:0.3"
Maybe it is better to give the real W:Co ratio, as e.g., 1, 2 or 1.3, or. 2.3, etc.
Section 3.2 - it is not clear how were the CNT synthesised - over differen catalyst (having different W:Co ratio)
Fig. 4 caption is incorrect - the SEM is presented in "b-f" while "a" presents Raman spectra.
Fig. 5 and 6 - a capital letter at the beggining of the caption is missing.
